# CLEAR: A Cost-Aware Routing System for Edge-Cloud Language Model Collaborative Inference

## Abstract

Large language models (LLMs) demonstrate exceptional performance across various natural language processing tasks but face significant computational and memory constraints, making direct deployment on resource-limited edge devices impractical. To address this challenge, we propose **CLEAR**, a cost-aware edge-cloud collaborative inference framework that efficiently integrates cloud-based LLMs with small language models (SLMs) running on edge devices. CLEAR introduces a cost-aware router that dynamically evaluates SLM-generated outputs and selectively routes low-quality output to cloud-based LLMs for refinement, balancing quality and computational efficiency. The framework incorporates two key innovations: KV cache management system and reinforcement learning-based router training. The KV cache management system prevents cache eviction and minimizes redundant computations by limiting concurrent cloud requests and optimizing retrieval efficiency. Additionally, the router is trained using reinforcement learning to make adaptive routing decisions that minimize cloud usage while maintaining output quality. Our experimental results demonstrate that CLEAR significantly reduces inference cost and latency while maintaining high response quality, outperforming existing cloud-edge collaborative inference methods. Specifically, it achieves a cost reduction of 46% while achieving similar performance or a performance improvement of 15% with similar inference cost. These findings highlight the potential of CLEAR as an efficient and scalable solution for real-time edge-cloud inference applications.

## 1 Introduction

Large Language Models (LLMs) have showcased remarkable capabilities across various natural language processing tasks. However, their superior performance comes with substantial computational and memory demands, making direct deployment on resource-constrained edge devices, such as smartphones and smartwatches, impractical. Typically, only smaller language models (SLMs) with reduced parameters can run efficiently on these devices, often resulting in suboptimal content quality. This challenge highlights the need for innovative approaches that harness the strengths of both LLMs and SLMs to achieve a balance between quality and efficiency.

To tackle this challenge, collaborative inference techniques between LLMs and SLMs have been proposed (Zhang et al., 2024; Yang et al., 2024) to leverage their complementary strengths, enhancing both efficiency and performance. However, existing cloud-edge collaborative inference systems (Jin & Wu, 2024; Hao et al., 2024; Zheng et al., 2025) often suffer from high latency, computational overhead, and increased communication costs due to transmitting full hidden states from the edge device to the cloud or requiring the cloud model to verify each output token, resulting in more cloud requests and higher overall latency.

To address these issues, during the inference, we aim to design a lightweight *router* to decide whether to request the cloud LLM or rely on the edge SLM to generate the next a few tokens. However, this approach introduces three key challenges:

- **How to maintain and re-use the KV cache across requests?** In collaborative inference, the same query may be sent to the cloud LLM multiple times as generation alternates between the cloud and

edge models. Excessive concurrent requests may result in previously generated KV cache entries being dropped or offloaded to CPU memory, leading to redundant generation or additional data transfers in subsequent cloud requests.

- **How to reduce communication costs and latency?** Each request to the cloud model introduces a fixed network communication delay, increasing the overall latency. To minimize this latency, the frequency of requests must be reduced to maintain system efficiency.

- **How to develop a cost-aware router to minimize resource costs while ensuring high-quality generation?** The router must balance maintaining high output quality with reducing cloud requests to minimize resource costs. This involves considering the fixed network communication cost, the computational cost of the LLM and SLM, and the output quality.

To address these challenges, we introduce **CLEAR** (**C**ost-aware **LLM E**dge-cloud **A**daptive **R**outing), which enables seamless collaboration between SLMs on edge devices and the LLM in the cloud. Specifically, a lightweight router on the edge device evaluates the quality of SLM outputs and forwards a request to the cloud LLM for re-generation if the quality is insufficient. Our framework tackles the above challenges through two key components: *KV cache management system* and *reinforcement learning-based router training*. To prevent KV cache eviction or swapping due to limited GPU memory, the KV cache management system restricts the number of simultaneous users and queues additional requests as needed. Additionally, request frequency is reduced by allowing the cloud LLM to generate multiple tokens for a single request. Meanwhile, we train the router using reinforcement learning (RL), where the reward function incorporates both output quality and generation costs. This enables the router to make efficient routing decisions between the edge SLM and the cloud LLM, optimizing for both communication and computational overhead. This holistic design allows CLEAR to achieve both cost-efficiency and high-quality generation.

The primary contribution of this paper is CLEAR, a collaborative inference framework that seamlessly coordinates SLMs on edge devices with the cloud-based LLM. Empirical results on four datasets demonstrate that CLEAR significantly reduces inference costs and latency while preserving output quality. Specifically, it achieves a 46% reduction in cost while maintaining comparable performance, or delivers up to a 15% performance improvement under similar inference costs. Furthermore, the results confirm the effectiveness of both the KV cache management system and the reinforcement learning-based router training. Collectively, these findings highlight the practicality and robustness of CLEAR for real-world edge-cloud deployment scenarios.

## 2 PRELIMINARIES

**Reinforcement Learning Framework for Collaborative Token-Level Inference.** We start by introducing the foundational concepts and notation for the Markov Decision Process for Token-Level Routing. In particular, the *state* is a sequence of tokens $\mathbf{s}_h = (x_0, \cdots x_m, y_0, \cdots, y_h) \in \mathcal{S}$, including both the input prompt $(x_0, \cdots x_m)$ and the response $(y_1, \cdots, y_h)$. At each time step $h$, the agent selects the SLM or LLM by choosing its action from $a_h \in \mathcal{A} = \{A_S, A_L\}$ according to its policy $\pi \in \mathcal{S} \mapsto \mathcal{A}$. The agent then proceeds to *next state* $\mathbf{s}_{h+1}$ by generating the next token as follows:

$$\mathbf{s}_{h+1} = \begin{cases} \mathbf{s}_h \oplus f_S(\mathbf{s}_h), \text{ if } a_h = A_S \\ \mathbf{s}_h \oplus f_L(\mathbf{s}_h), \text{ if } a_h = A_L \end{cases}, \tag{1}$$

which suggests that next token will be generated by the large language model $f_L$ when agent selects $A_L$ and by the small language model $f_S$ when the agent selects $A_S$. For each step $h$, the agent receives the reward $\bar{r}(\mathbf{s}, a) = r(\mathbf{s}) - c(\mathbf{s}, a)$ where $r(\mathbf{s})$ indicates the quality of the response and $c(\mathbf{s}, a)$ indicates the cost of calling different language models. The $Q$-function is defined as a discounted infinite horizon manner with discounted factor $\gamma$

$$Q^\pi(\mathbf{s}, a) = \mathbb{E}\left[\sum_{t=0}^\infty \gamma^t \bar{r}(\mathbf{s}_t, a_t) | \mathbf{s}_0 = \mathbf{s}, a_0 = a, \pi\right],$$

the value function $V^\pi(\mathbf{s})$ and the advantage function $A^\pi(\mathbf{s})$ is defined as

$$V_h^\pi(\mathbf{s}) = \mathbb{E}_{a \sim \pi(\cdot|\mathbf{s})}[Q_h^\pi(\mathbf{s}, a)], A^\pi(\mathbf{s}) = Q^\pi(\mathbf{s}) - V^\pi(\mathbf{s}).$$

The routing policy optimization aims to maximize the $Q$ function where the optimal $Q$ function and value function are defined by $Q^*(\mathbf{s}, a) = \max_\pi Q^\pi(\mathbf{s}, a)$ and $V^*(\mathbf{s}) = \max_\pi V^*(\mathbf{s})$ accordingly.

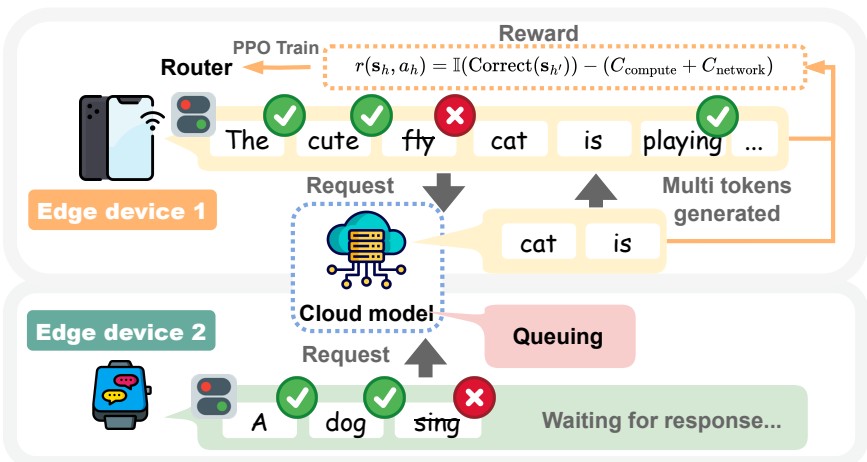

Figure 1: The overview of CLEAR. The edge model coupled with a cost-aware router is deployed on each edge device. The router is trained as a RL problem with the reward considering both response accuracy and generation cost. Then, the router is leveraged to perform Cloud-Edge collaborative inference by determining whether to request the cloud model for generation refinement. A KV cache management system is implemented on the cloud to ensure KV cache reuse and reduce generation latency by queuing extensive requests and generating multiple tokens in one request.

**GPU Memory Management in LLM Systems.** Modern machine learning systems require new strategies to manage GPU memory and increase performance. Given a sequence of tokens $\mathbf{s}_t = (x_0, x_1, \ldots, x_t)$, the KV cache stores the key and value embeddings $\mathbf{k}_i, \boldsymbol{b}_i$ for all tokens $x_i \in \mathbf{s}_t$. By retaining these embeddings, the system can substantially reduce the computation cost needed to generate subsequent tokens $x_{t+1}, x_{t+2}, \cdots$. However, storing a massive single-token-level KV cache in GPU memory can lead to memory fragmentation. To address this issue, Kwon et al. (2023) introduced *Paged Attention*, drawing inspiration from the concept of *virtual memory with paging* (Kilburn et al., 1962) in operating systems. Specifically, PagedAttention divides GPU memory into multiple fixed-length 'blocks', each capable of storing a fixed number $B$ of KV cache entries $\{(\mathbf{k}_1, \boldsymbol{b}_1), \cdots, (\mathbf{k}_B, \boldsymbol{b}_B)\}$.

## 3    COST-AWARE EDGE-CLOUD COLLABORATIVE COMPUTING

In this section, we describe our proposed **CLEAR**, a cost-aware LLM edge-cloud adaptive routing framework for collaborative computing. As illustrated in Figure 1, each edge device is equipped with a local edge model which is efficient but potentially inaccurate. A powerful but computationally expensive cloud model is deployed on cloud. Each edge client learns its own *routing policy* $\pi(\cdot|\mathbf{s}_h)$ which determines whether request the cloud model to generate the next several tokens. To enable efficient collaboration between the edge model and the cloud model, the proposed framework comprises two core components: the **KV cache management system** and **cost-aware routing policy**. Specifically, on the cloud side, clients may repeatedly request the cloud model for help during a single local query. To improve the efficiency of the cloud model response, we propose a **KV cache management system** to ensure that the KV cache from previous requests for the same query is properly retained and reused across subsequent requests. In addition, on the edge side, we implement a **cost-aware routing policy** to balance the quality and the cost (speed) of the generation.

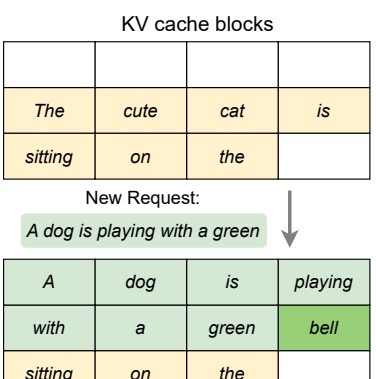

Figure 2: The cloud model receives a new request after just finishing the old one. The previous KV cache must be dropped when processing the new request.

Unlike previous works (Zheng et al., 2025; Shen et al., 2024) that focus on single-token routing, we

---

**Algorithm 1** The KV Cache Management System for Cloud Inference

---

1: **Input:** Cloud model $M_L$, Edge models $M_S$, Server capacity $C$
2: **Initialize:** Running query ids $S_q = \phi$, Waiting queue $Q_w = \phi$, Completion request $R^C$
3: **while** Not Terminated **do**
4:     Receive new requests $\{R_i^h\}$ from edge devices $M_S$.
5:     **for** each request $R_i^h$ **do**
6:         **if** $R_i^h$ is completion request $R^C$ **then**
7:             Set $S_q = S_q \setminus \{i\}$
8:         **else if** $i \in S_q$ or $|S_q| < C$ **then**
9:             Set $S_q = S_q \cup \{i\}$
10:            Send multiple token request $R_i^h$ to the cloud model $M_L$.
11:        **else**
12:            Append $R_i^h$ to $Q_w$.
13:    Receive the completed request $\{R_i^h\}$ with multiple tokens from the cloud model $M_L$.
14:    **for** each request $R_i^h$ **do**
15:        **if** $R_i^h$ is completion request $R^C$ **then**
16:            Set $S_q = S_q \setminus \{i\}$
17:            **if** $Q_w \neq \phi$ **then**
18:                Pop a new request $R_i^h$ from the waiting queue $Q_w$
19:                Send multiple token request $R_i^h$ to the cloud model $M_L$.
20:                Set $S_q = S_q \cup \{i\}$
21:    Send the request result of $R_i^h$ to the edge device $M_S$.

---

extended the routing policy that allows the cloud model to request a variable length of tokens at the same time to improve the throughput of the cloud model.

## 3.1 CLOUD MODEL: KV CACHE MANAGEMENT SYSTEM

On the cloud side, the model receives a request $\mathbf{s}_h^i = (\mathbf{x}^i, \mathbf{y}_h^i)$ from the $i$-th client, consisting of the input prompt $\mathbf{x}^i$ and the first $h$ tokens of the response. To accommodate future requests $\mathbf{s}_{h'}^i$ with extended responses ($h' > h$) for the same prompt, the KV cache must be preserved so that previously computed states can be reused instead of recomputed. Beyond computation, each request also incurs a fixed network communication delay, which further limits generation throughput. To mitigate these issues, we propose a KV cache management system for the cloud model, featuring two key mechanisms: (1) a query queuing mechanism and (2) a multi-token generation mechanism, designed to improve cache utilization and reduce latency.

**Query Queuing Mechanism.** Excessive concurrent requests can lead to KV cache eviction or swapping to CPU memory due to limited GPU resources on the cloud. Under that situation, when subsequent requests of the same query $\mathbf{s}_{h'}^i$ are proposed, redundant computations or additional data transfers are required for these subsequent requests, leading to suboptimal performance of the whole inference system. For example, as demonstrated in Figure 2, a new request $R_2$ arrives at the cloud model after the cloud model has just finished the previous request $R_1$. The generation process of request $R_2$ requires evicting the KV cache of $R_1$ due to insufficient GPU memory, which may lead to redundant computation in the subsequent request of $R_1$.

To address this challenge, we limit the number of simultaneous queries handled by the cloud model. When the maximum server capacity $C$ is reached, new queries are queued until prior collaborative inference processes are completed. Concretely, we implement a queueing mechanism with a bounded running queue $S_q$ of length $C$, and an unbounded waiting queue $Q_w$. Only requests in $S_q$ are allowed to start the generation on the cloud. A new request is placed in $S_q$ if space is available; otherwise, it is appended to $Q_w$. As requests in $S_q$ complete, new requests from $Q_w$ are admitted into $S_q$. Completion is defined by either the generation of an <EOS> token or reaching the maximum token limit at the edge or cloud. This queuing mechanism, detailed in Algorithm 1, ensures that the KV cache for active queries remains in GPU memory, enabling efficient reuse across multiple cloud requests and reducing redundant computation and additional data transitions.

---

**Algorithm 2** Router Policy Optimization for Edge SLM

---

1: **Input:** Max generation length $T$, exploration probability $p$
2: **Initialize:** Policy network $\pi_{\boldsymbol{\theta}}$ with random weights. Replay buffer $\mathcal{D} = \phi$.
3: Perform warm-start supervised training for $\pi_{\boldsymbol{\theta}}$ on pre-collected routing data.
4: **while** Not Converged **do**
5:     **for** each generation episode and each input query $\mathbf{x}$ **do**
6:         Initialize state $\mathbf{s}_0 = (\mathbf{x}, \text{empty response})$, step $h = 0$.
7:         **while** not $\texttt{<EOS>}$ and $h < T$ **do**
8:             Sample action $a_h \sim \pi_{\boldsymbol{\theta}}(a|\mathbf{s}_h)$ and flip it with probability $p$ for exploration.
9:             **if** $a_h = A_L$ **then**                   ▷ Request Cloud Model
10:                 Determine number of tokens $N_h$ to generate to fill the current KV block.
11:                 Set penalty reward $r_h = -\lambda \frac{B+N_h}{T}$.
12:             **else** Set $N_h = 1, r_h = 0$.                 ▷ Use Edge Model
13:             Generate $N_h$ tokens $\Delta\mathbf{y}$ and next state $\mathbf{s}_{h'} = \mathbf{s}_h \oplus \Delta\mathbf{y}$.
14:             **if** $\texttt{<EOS>}$ is generated in $\Delta\mathbf{y}$ **then** Set reward $r_h \leftarrow r_h + \mathbb{1}[\text{Correct}(\mathbf{s}_{h'})]$.
15:             Store transition $(\mathbf{s}_h, a_h, r_h, \mathbf{s}_{h'})$ in buffer $\mathcal{D}$
16:             Update step counter $h \leftarrow h + N_h$.
17:     Update policy $\pi_{\boldsymbol{\theta}}$ via PPO with replay buffer $\mathcal{D}$.
18:     Clear replay buffer $\mathcal{D}$; Linearly decay probability $p$.

---

**Multi-Token Generation Mechanism.** Though the Query Queuing Mechanism allows KV caches from previous requests to be retained and reused, each cloud request still incurs a fixed network delay. To mitigate this latency, we reduce the number of requests by enabling the cloud model to generate multiple tokens at once. However, PagedAttention (Kwon et al., 2023), widely used in modern inference sys-

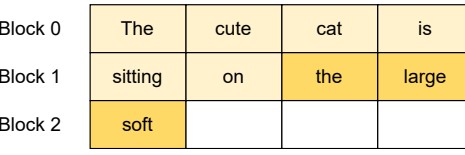

Figure 3: The KV cache blocks after the cloud model has just finished a request.

tems (Ye et al., 2025), has a limitation: only fully filled KV cache blocks can be reused across requests, while partially filled blocks, typically the last block of a generation, must be recomputed. For example, Figure 3 shows that the first eight tokens form two full blocks (Block 0 and Block 1), which are reusable, but the final token only partially fills Block 2, rendering it unreusable and forcing redundant computation in subsequent requests. This occurs because request $R_1^h$ terminates before completing an entire block. To address this inefficiency, we dynamically determine the number of tokens to generate based on the available slots in the current block. Generation halts once a block is completely filled or the collaborative inference ends. This strategy maximizes KV cache reuse and avoids redundant computation.

## 3.2 EDGE SIDE: ROUTER POLICY OPTIMIZATION

Given the KV cache management system for efficient implementation on the cloud side, the edge side needs to determine whether the cloud model should be requested based on both the generation cost, including the fixed network communication delay and the computational cost proportional to the number of tokens generated, and the quality improvement introduced by each request. In addition, since we allow the cloud model to generate multiple tokens in a single request, the quality improvement introduced by each request becomes difficult to evaluate. To address this challenge, we extend the RL framework for token-level routing discussed in Section 2 to accommodate this multistep generation process. In detail, different from the state transition described in equation 1, when choosing action $A_L$ requesting the cloud model, the next state is given by

$$\mathbf{s}_{h'} = \mathbf{s}_h \oplus f_L(\mathbf{s}_h) \oplus f_L(\mathbf{s}_{h+1}) \cdots \oplus f_L(\mathbf{s}_{h'}), \tag{2}$$

which takes multiple tokens from the cloud model $f_L$.

The reward function $r(\mathbf{s}_h, a_h)$ is defined as follows: First, when equation 2 completes the generation with $\texttt{<EOS>}$, we assign $r(\mathbf{s}_h, a_h)$ as the correctness of the response. Otherwise, when the generation does not complete, we assign a penalty reward for calling the cloud model. As a result, the reward

Table 1: The statistics of our evaluation datasets.

| Dataset | Domain | Task | # choices | Train size | Test size |
|---------|--------|------|-----------|------------|-----------|
| Commonsense QA | General | CoT + Multi-choice | 5 | 9,741 | 1,221 |
| ARC-Challenge | Reasoning | CoT + Multi-choice | 4 | 1,119 | 299 |
| OpenBookQA | Reasoning | CoT + Multi-choice | 4 | 4,957 | 500 |
| MATH | Math | Question answering | N/A | 7,500 | 5,000 |

assignment can be written as follows:

$$r(\mathbf{s}_h, a_h) = \begin{cases} \mathbb{1}[\text{Correct}(\mathbf{s}_{h'})] \text{ if } \texttt{<EOS>} \text{ is encountered} \\ -\lambda \frac{B+N_h}{T} \mathbb{1}[a_h = A_L] \text{ otherwise} \end{cases}$$

where $B$ is the KV cache block size, $N_h$ is the number of tokens the cloud model needs to generate in current request, $T$ is the limitation to the maximum number of generated tokens of the whole generation process. We leverage $B/T$ to represent the fixed network latency penalty and $N_h/T$ to represent the computational cost of the cloud model. $\lambda$ is a hyperparameter to balance the trade-off between the generation cost and the quality improvement. The reward definition ensure the router is trained to minimize the generation cost while ensuring high-quality outputs.

Given the reward function, we employ the Proximal Policy Optimization (PPO) (Schulman et al., 2017) to learn the value function $V_{\phi}$ and the policy $\pi_{\theta}$ by minimizing the following two losses:

$$\mathcal{L}_{\text{PPO}}(\boldsymbol{\theta}) = -\mathbb{E}_t[\min\{r_h(\boldsymbol{\theta})A_h, [r_h(\boldsymbol{\theta})]_{1-\epsilon}^{1+\epsilon}A_h\}]$$

$$\mathcal{L}_V(\boldsymbol{\phi}) = \mathbb{E}[(V_{\phi}(\mathbf{s}_h) - V_h^{\text{tar}}(\mathbf{s}_h))^2],$$

where the expectation is taken over the data sampled by the policy used in the previous round $\boldsymbol{\theta}_{\text{old}}$. The probability ratio $r_h(\boldsymbol{\theta})$ is defined as $r_t(\boldsymbol{\theta}) = \pi_{\boldsymbol{\theta}}(a_h|\mathbf{s}_h)/\pi_{\boldsymbol{\theta}_{\text{old}}}(a_h|\mathbf{s}_h)$. The target value function is defined as $V_h^{\text{tar}}(\mathbf{s}_h) = r(\mathbf{s}_h, a_h) + \gamma V_{\phi}^{\perp}(\mathbf{s}_{h+1})$ with $V^{\perp}$ stopping the propagation of the gradient backward. The advantage function is defined by $A_h = r(\mathbf{s}_h, a_h) + \gamma V_{\phi}^{\perp}(\mathbf{s}_{h+1})$. To encourage the exploration for the policy optimization, we random flip the actions with a probability of $p$ during the sampling. The probability $p$ is a hyperparameter controlling the exploration rate, which is linearly decreased to 0 during the training process. We also perform a warm-start training following previous token-level routing methods (Jin & Wu, 2024; Zheng et al., 2025), where the initial policy model is trained using supervised learning with pre-collected token-level routing information. The whole process is also illustrated in Algorithm 2.

## 4 EXPERIMENTS

In this section, we evaluate the performance of CLEAR aiming to answer the following questions: (1) Compared with the previous cloud-edge collaborative inference framework, how does our framework perform in terms of the inference time and the quality of the generated response? (2) How the KV cache management system we proposed, including the query queuing mechanism and the multi-token generation mechanism improve the overall throughput of our system? (3) Does the RL training process of our verifer fill in the verifer with the knowledge of making intelligent routing decisions based on both the generation cost and the output quality? Does the RL training outperforms the supervised learning style?

### 4.1 EXPERIMENTAL SETUP

**Dataset Description.** We evaluate CLEAR and baseline models on four commonly used benchmark datasets. The Commonsense QA dataset (Talmor et al., 2019) comprises 12,102 questions that require various types of commonsense knowledge to answer. The ARC-Challenge dataset (Clark et al., 2018) contains 1,418 authentic, grade-school-level multiple-choice science questions. The OpenBookQA dataset (Mihaylov et al., 2018) includes 5,957 questions designed to be answered using a provided set of elementary science facts combined with broad common knowledge. Lastly, the MATH dataset (Hendrycks et al., 2021) includes 12.5K mathematical problems sourced from competitions. Table 1 provides the statistical details of these datasets and detailed descriptions of these datasets are in Appendix B.

**Baselines.** We compare CLEAR with CE-CoLLM (Jin & Wu, 2024) and hybrid SLM-LLM models (Hao et al., 2024) (Hym). CE-CoLLM utilizes confidence scores to determine whether to

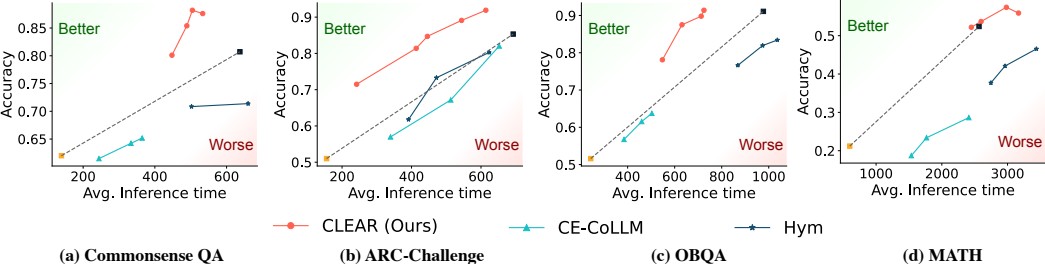

(a) Commonsense QA  (b) ARC-Challenge  (c) OBQA  (d) MATH

Figure 4: The accuracy vs computational costs curve of CLEAR and our baselines. The yellow and grey squares represent the performance of slm and llm respectively. The grey line represents the random routing strategy. Points closer to the top-left corner indicate better acceleration performance.

request the cloud LLM for the next token, transmitting hidden states from the edge to the cloud during the process. Hym, on the other hand, allow the edge model to generate candidate tokens, which are then verified by the cloud model for final generation. Since neither of these has publicly available code, we implement them based on the descriptions provided in the original papers.

**Evaluation Plan.** We evaluate the performance of CLEAR and baseline models using the test set of each dataset and their respective evaluation metrics. Specifically, for each approach, we apply a threshold $\tau$ to balance the trade-off between response accuracy and average inference time. The average inference time is calculated as the total time taken to generate responses for all queries across all edge devices, divided by the total number of queries. To illustrate the performance of CLEAR and the baselines, we plot the accuracy curve against average inference time.

**Implementation Details.** Our framework is implemented using the Hugging Face Transformers library (Wolf et al., 2020) for the edge SLM and the vllm library for the centralized cloud LLM. For both the SLM and LLM, we employ the Qwen3 series Yang et al. (2025), where Qwen3-1.5b serves as the edge SLM and Qwen3-72b as the cloud LLM. The router is designed as a multilayer perceptron (MLP) network, consisting of three hidden layers with ReLU activation (Fred & Agarap, 2018), BatchNorm normalization (Ioffe & Szegedy, 2015), and a dropout rate of 0.1. Initially, it undergoes training via the behavior cloning strategy, followed by fine-tuning using the Proximal Policy Optimization (PPO) algorithm (Schulman et al., 2017). The value net employed in the PPO algorithm is a similar MLP network as the router, only with different final output dimensions. We deploy the cloud model on a single node with 8 NVIDIA H100 GPUs, while the edge model is deployed on another node with 2 to 8 NVIDIA H100 GPUs, stimulating multiple edge devices.

## 4.2 OVERALL PERFORMANCE

We conduct extensive experiments to evaluate the performance of CLEAR across all datasets, and the results are presented in Figure 4. The findings can be summarized in two key points. First, CLEAR consistently outperforms all baseline models and a random routing strategy on every dataset. Specifically, CLEAR achieves up to 15% higher accuracy than our baselines for a similar inference time, or reduces inference time by up to 46% while maintaining similar performance. Second, on challenging datasets including Commonsense QA, ARC-Challenge, and MATH, our method's performance is so effective that it even surpasses the accuracy of using the powerful cloud-based LLM alone, with improvements of up to 7%.

The outperformance of CLEAR over other baselines is due to our superior token-level routing mechanism. Unlike methods such as CE-CoLLM, which transmit full hidden states from the edge to the cloud, CLEAR only sends token IDs. This design drastically reduces the volume of data transmitted and the associated communication latency. Furthermore, compared to hybrid models that often rely on speculative decoding, our token routing is more flexible. It does not require every token generated by the edge model to be verified by the cloud LLM, thus avoiding the frequent, high-latency cloud requests that can become a bottleneck in edge-cloud collaboration.

The ability of CLEAR to surpass the standalone cloud LLM is attributed to the fact that we use reinforcement learning (RL) to optimize our router The RL process trains the router not just to reduce cost, but to identify the specific strengths of each model. It learns which parts of a task can be efficiently and accurately handled by the SLM and, more importantly, which critical tokens require the LLM's advanced reasoning capabilities. By dynamically leveraging the complementary advantages of both models, CLEAR achieves a synergistic performance that can even surpass that of the more powerful LLM operating alone.

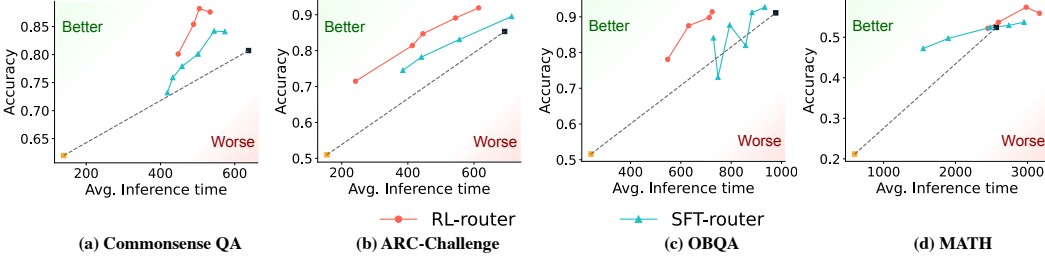

|   |   |   |   |
|---|---|---|---|
| (a) Commonsense QA | (b) ARC-Challenge | (c) OBQA | (d) MATH |

Figure 6: The accuracy vs computational costs curve of CLEAR with RL-based router and SFT-based router. The yellow and grey squares represent the performance of slm and llm respectively. The grey line represents the random routing strategy. Points closer to the top-left corner indicate better acceleration performance.

## 4.3 ANALYSIS OF KV CACHE MANAGEMENT SYSTEM

In this section, we conduct analyses on our query queuing and multi-token generation mechanisms.

**Analysis of Query Queuing Mechanism.** To assess the effectiveness of the query queuing mechanism, we vary the number of simultaneous requests to simulate different server capacities and measure system throughput. Specifically, we continuously submit generation requests to the cloud model and record the number of tokens generated per second as the throughput metric. The experiments compare the cloud server's performance with and without the query queuing mechanism. In addition, we deploy our cloud model on two H100 GPUs instead of eight, thereby limiting available GPU memory and maximum concurrency to more easily showcase the effectiveness of the query queuing mechanism. The results, shown in Figure 5, reveal a clear trend: initially, system throughput improves as server capacity increases, enabling greater query concurrency and better GPU utilization. However, beyond a

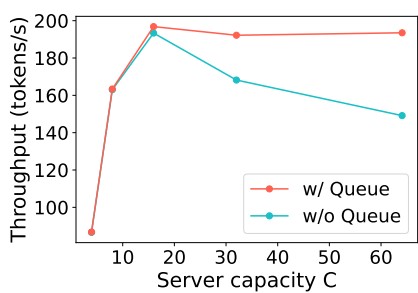

Figure 5: The throughput of our cloud model with and without the query queuing generation mechanism.

certain threshold, the absence of the query queuing mechanism leads to a decline in throughput due to GPU memory exhaustion. This results in KV cache entries being evicted or swapped to CPU memory, causing redundant generation and additional data transfers in subsequent requests. In contrast, with the query queuing mechanism, throughput remains stable, ensuring efficient resource utilization and high performance. These findings highlight the mechanism's critical role in maintaining optimal system performance.

**Analysis of Multi-Token Generation Mechanism.** We also evaluate the impact of the multi-token generation mechanism on the throughput of our cloud model. To assess its effectiveness, we first measure the model's throughput without the mechanism. Additionally, with the mechanism enabled, we vary the KV cache block size to analyze throughput under different numbers of tokens generated per request.

Table 2: Throughput comparison at different KV cache block sizes (in tokens).

| KV size | 1 | 8 | 16 | 32 |
|---|---|---|---|---|
| Throughput | 87.96 | 215.48 | 240.25 | 259.40 |

The results, shown in Table 2, confirm that the multi-token generation mechanism significantly enhances throughput, achieving higher efficiency as more tokens are generated per request. This aligns with our expectations, as the mechanism reduces the number of requests sent to the cloud model, minimizing communication overhead and computational costs while improving overall system efficiency.

## 4.4 ANALYSIS OF REINFORCEMENT LEARNING-BASED ROUTER TRAINING

Finally, we compare the performance of the router trained by reinforcement learning with that trained by supervised learning. The results, presented in Figure 6, show that the router trained with the reinforcement learning algorithm significantly outperforms the supervised learning-based router in all the simple multi-choice datasets. Especially, on the ARC-Challenge dataset, CLEAR with the RL-based router achieves similar accuracy with up to 4x less computational cost, compared to the SFT-based version. In addition, for these complex math datasets, the RL-based router consistently chooses the cloud model, which is the best strategy under that situation, regardless of the threshold,

while the SFT-based router does not. This demonstrates the effectiveness of reinforcement learning in training the router to make intelligent routing decisions based on both generation cost and output quality, ensuring optimal performance of the collaborative inference process.

## 5 RELATED WORK

In this section, we review prior related works to our framework. We first discuss cloud-edge computing methods, followed by collaborative inference frameworks. Finally, we introduce other LLM inference acceleration methods beyond this collaborative inference paradigm.

**Cloud-Edge Computing Method.** The cloud-edge computing paradigm has emerged as a key solution for improving computational efficiency, reducing latency, and enhancing data privacy (Chen & Li, 2025; Wang & Singh, 2025; Lv et al., 2025; Ji et al., 2025). With the advancement of LLMs, Cloud-Edge LM collaboration has gained increasing attention, with recent frameworks focusing on optimizing distributed inference. For instance, EdgeShard (Zhang et al., 2024) partitions LLM inference between cloud and edge; PerLLM (Yang et al., 2024) optimizes scheduling for efficiency and performance; CE-CoLLM (Jin & Wu, 2024) processes initial layers on the edge and dynamically offloads based on confidence scores. Hao et al. (2024) extends speculative decoding (Leviathan et al., 2023) by using a small edge-based model for token proposals, with a cloud LLM as a verifier. However, these methods often introduce significantly additional latency and communication overhead, leading to suboptimal performance in real-time applications.

**Collaborative Inference.** Collaborative inference frameworks (Han et al., 2024; Mohammadshahi et al., 2024; Davis & Miller, 2025) seek to harness the complementary strengths of both small and large models to improve efficiency and performance. Recent advancements in this area have introduced various routing strategies and model adaptation techniques (Jang et al., 2023; Lu et al., 2023; Wang et al., 2024). Among these, RouteLLM (Ong et al., 2024) frames the routing process as a classification task, letting either LLM or SLM generate the whole results. Recently, Co-LLM (Shen et al., 2024) and CITER (Zheng et al., 2025) introduced token-level routers that determine routing for each token, directing them to either an SLM or an LLM based on complexity to optimize computational resource allocation. Building on this, AdaRoute (Lee & Park, 2025) integrates mixture-of-experts (MoE) principles to create more dynamic and fine-grained routing policies. However, these frameworks primarily focus on routing strategies and do not address the challenges of cloud-edge collaboration, where a single cloud LLM serves multiple edge SLMs. Routing System (She et al., 2025) builds a serving system that supports deploying cloud-based collaborative inference systems in the real world. Efficient resource management is crucial in this setting to maintain acceptable throughput.

**Alternative Methods for LLM Inference Acceleration.** Beyond the previously discussed techniques, various methods have been developed to improve LLM inference efficiency (Kwon et al., 2023; Chen et al., 2024a; Xu et al., 2025; Qinsi et al., 2025; Zhao & Kumar, 2025). A widely studied approach, Speculative Decoding (Leviathan et al., 2023; Chen et al., 2023), employs a small draft model to propose tokens, with a larger LLM verifying or rejecting them. Expanding on this, Speculative Streaming (Bhendawade et al., 2024) accelerates inference by predicting n-grams per forward pass instead of single tokens by redesigning the LLM architecture. Medusa (Cai et al., 2024) further optimizes this process by introducing auxiliary prediction heads for lightweight n-gram generation. Recent advancements such as Hydra (Liu & Zhang, 2025) extend this multi-head approach to generate and verify entire token trees in parallel. Other methods, such as SpecInfer and Sequoia (Miao et al., 2023; Chen et al., 2024b), utilize tree-based parallelism to enhance decoding and verification, further accelerating the inference process.

## 6 CONCLUSION

In this work, we propose a novel cost-aware edge-cloud collaborative inference framework, CLEAR, that leverages a router on the edge device to determine whether the cloud model should regenerate the current tokens. Our framework comprises a KV cache management system on the cloud model and a router on the edge device. The KV cache management system ensures the KV cache from previous requests for the same query is retained and reused across subsequent requests, optimizing latency and throughput. The router assesses the quality of the edge model's outputs, minimizing redundant computations and optimizing resource utilization. Extensive experiments demonstrate that CLEAR outperforms baseline methods in terms of accuracy and computational efficiency, highlighting the effectiveness of our framework in cloud-edge collaborative inference tasks.

ETHICS STATEMENT

The authors of this paper have read and adhered to the ICLR Code of Ethics. Our work introduces CLEAR, a cost-aware edge-cloud collaborative inference framework designed to improve the efficiency of large language models. The primary goal of this research is to reduce the computational and financial costs associated with LLM inference, making these powerful tools more accessible and sustainable. We believe this work has a positive societal impact by democratizing access to state-of-the-art AI.

We have taken care to ensure that our research is conducted responsibly. All datasets used for training and evaluation are publicly available benchmarks for natural language processing and do not contain personally identifiable information (PII) or other sensitive data. The framework itself is designed to be a general-purpose tool and does not inherently encourage or facilitate harmful applications. We acknowledge that, like any powerful technology, LLMs can be misused. However, the focus of our work is on the efficiency of the underlying system, not on the generation of content. We encourage the community to continue to explore and implement safeguards against the misuse of language models.

REPRODUCIBILITY STATEMENT

To ensure the reproducibility of our research, we will make all relevant artifacts publicly available upon the publication of this paper. This will include the complete source code for the CLEAR framework, including the implementation of the KV cache management system and the reinforcement learning-based router. We will also provide the exact versions of the datasets used in our experiments, along with the scripts for data preprocessing and evaluation.

A detailed description of our methodology is provided in Section 3, which includes the algorithms for the query queuing mechanism (Algorithm 1) and the router policy optimization (Algorithm 2). The experimental setup, including the specific models used (Qwen3-1.5b and Qwen3-72b), the hardware configuration, and the evaluation metrics, is described in Section 4.1. Further details, including the prompts used for each dataset, are available in the Appendix B. We believe these resources will be sufficient to allow for the full replication of our results.

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

## A  LLM USAGE FOR PAPER WRITING

We have used Large Language Models (LLMs) for the limited purpose of language polishing. The core aspects of this paper, including the initial ideation, structural framework, and primary writing, were completed entirely by human authors. All text polished by LLMs was subsequently reviewed, edited, and, when necessary, rewritten by the human authors to ensure the accuracy and integrity of the content. The human authors are fully responsible for the research design, experiments, analysis, and the final version of this work.

## B  DATASET DESCRIPTION

In this section, we describe our benchmark datasets with more details.

### B.1  COMMONSENSE QA

CommonsenseQA is a substantial multiple-choice dataset crafted to assess a model's proficiency in commonsense reasoning. It comprises a total of 12,102 questions, each structured with one correct response and four plausible but incorrect alternatives. To succeed, models must rely on an extensive range of commonsense knowledge to differentiate the correct choice from distractors. Unlike conventional QA datasets, CommonsenseQA places a strong emphasis on diverse reasoning paradigms, incorporating causal relationships, temporal sequences, and conceptual associations, thereby necessitating a deeper understanding of everyday knowledge.

### B.2  ARC-CHALLENGE

The AI2 ARC dataset is an extensive compilation of 7,787 multiple-choice science questions at the grade-school level, carefully designed to drive progress in automated question-answering. It is systematically categorized into two parts: the ARC-Easy Set and the ARC-Challenge Set. Our study focuses on the ARC-Challenge Set, a subset intentionally constructed to include particularly challenging questions. These questions were selected based on their failure cases under both a standard retrieval-based approach and a word co-occurrence method, ensuring that they pose a significant challenge to models. The ARC-Challenge Set acts as a rigorous benchmark, demanding models to go beyond simple pattern recognition and apply deeper scientific reasoning.

### B.3 OPENBOOKQA

OpenBookQA is a question-answering dataset designed to emulate open-book exams, assessing a model's ability to reason with external knowledge. It contains 5,957 multiple-choice questions focused on elementary-level science. Each question is accompanied by a small "open book" of 1,326 core science facts. Crucially, answering the questions requires multi-step reasoning that combines a fact from the provided open book with broad, external common knowledge not contained within it. For example, a model might need to use the open-book fact "metals conduct electricity" along with the common knowledge that "a suit of armor is made of metal" to answer a question. This design ensures that models cannot succeed through simple information retrieval alone but must instead demonstrate a deeper understanding of the subject matter and its application in novel contexts, making it a challenging benchmark for advanced reasoning.

### B.4 MATH

The Mathematics Aptitude Test of Heuristics (MATH) dataset is a comprehensive collection of 12,500 challenging mathematics problems sourced from various competitions, including the AMC 10, AMC 12, and AIME Hendrycks et al. (2021). Each problem is accompanied by a detailed, step-by-step solution, enabling models to learn the process of deriving answers and providing explanations. The dataset encompasses a wide range of mathematical disciplines, such as: Prealgebra, Algebra, Number Theory, Counting and Probability, Geometry, Intermediate Algebra, Precalculus. Problems are categorized by difficulty levels, ranging from 1 to 5, allowing for a nuanced assessment of a model's problem-solving abilities across various complexities. The solutions are meticulously formatted in LaTeX, ensuring clarity and consistency in mathematical notation. A distinctive feature of the MATH dataset is its focus on problems that require more than straightforward application of formulas; they often demand creative problem-solving techniques and heuristics. This design makes the dataset particularly suitable for training and evaluating models on complex mathematical reasoning tasks.

## C PROMPT

In this section, we illustrate the prompt we used for each dataset.

### C.1 MULTIPLE-CHOICE QUESTION PROMPT

For datasets containing multiple-choice questions, such as Commonsense QA and ARC-Challenge, we employ a structured prompting approach to guide large language models (LLMs). This prompt is specifically designed to elicit both a detailed explanation of the reasoning process and a clearly formatted final answer, ensuring consistency in response generation.

---

**Multiple-Choice Question Prompt**

```
Question:
$question
Choices:
(A). $choice_A
(B). $choice_B
(C). $choice_C
(D). $choice_D
Please format your response in the following way:
[Explanation]. Therefore, the answer is answer (label).
Ensure the final sentence includes the answer followed by the
label in parentheses.
Answer and Reasoning: $response
```

---

## C.2 MATH PROBLEM PROMPT

To ensure structured reasoning and clear presentation, mathematical problems from the MATH datasets are presented to large language models (LLMs) using a specialized prompt. This prompt instructs the model to generate a detailed, step-by-step solution formatted in LaTeX, emphasizing logical progression. Additionally, the final numerical or symbolic answer should be formatted following the answer format of each dataset, enhancing readability and standardizing response formatting.

---

**Math Problem Prompt**

```
Solve the following math problem step by step. Ensure all
reasoning is shown clearly in LaTeXformat, and the final answer
should be enclosed using the LaTeX command \boxed{} to display
it in abox.
Question: \$question
Solution: \$solution
```

---

## D EXAMPLES

This section provides representative examples from each dataset, showcasing the corresponding questions, the prompts utilized, and the responses generated by large language models (LLMs). By presenting these instances, we aim to illustrate the structure of our prompts and clarify the specific nature of the tasks associated with each dataset.

### D.1 COMMONSENSE QA

---

**Commonsense QA Example**

```
Question: A revolving door is convenient for two direction
travel, but it also
serves as a security measure at a what?
Choices:
A. bank
B. library
C. department store
D. mall
E. new york
Please format your response in the following way:
[Explanation]. Therefore, the answer is answer (label).
Ensure the final sentence includes the answer followed by the
label in parentheses.
Answer and Reasoning: 1. Analyze the choices: We need to find
the place where a revolving door serves as a security measure
for two-direction travel.\n2. Eliminate irrelevant choices: (B)
library, (D) mall, and (E) New York are not specific enough or
not related to security measures.\n3. Compare the remaining
choices: (A) bank and (C) department store both have security
concerns, but banks typically have more stringent security
measures due to the handling of money and valuable assets.\n 4.
Choose the most appropriate answer: (A) bank, as it is the
place where a revolving door serves as a security measure for
two-direction travel.\n\n Therefore, the answer is (A).
```

---

## D.2 ARC-CHALLENGE

---

**ARC-Challenge Example**

```
Question: An astronomer observes that a planet rotates faster
after a meteorite impact. Which is the most likely effect of
this increase in rotation?
Choices:
A. Planetary density will decrease.
B. Planetary years will become longer.
C. Planetary days will become shorter.
D. Planetary gravity will become stronger.
Please format your response in the following way:
[Explanation]. Therefore, the answer is answer (label).
Ensure the final sentence includes the answer followed by the
label in parentheses.
Answer and Reasoning: The astronomer observes that a planet
rotates faster after a meteorite impact. This means that the
planet is now completing one full rotation on its axis in a
shorter amount of time than it did before the impact.
(A). Planetary density will decrease. This is not necessarily
true. The density of a planet is determined by its mass and
volume, not its rotation speed.
(B). Planetary years will become longer. This is also not
necessarily true. The length of a year is determined by the
time it takes for a planet to complete one orbit around its
star, not its rotation speed.
(C). Planetary days will become shorter. This is the most
likely effect of the increase in rotation. Since the planet is
now rotating faster, it will take less time for one full
rotation to occur, resulting in shorter days.
(D). Planetary gravity will become stronger. This is not
necessarily true. The strength of a planet's gravity is
determined by its mass and distance from its star, not its
rotation speed.
Therefore, the answer is (C) Planetary days will become shorter.
```

---

## D.3 OPENBOOK QA

---

**OpenBook QA Example**

```
Question: A person wants to start saving money so that they can
afford a nice vacation at the end of the year. After looking
over their budget and expenses, they decide the best way to
save money is to
Choices:
A. make more phone calls.
B. quit eating lunch out.
C. buy less with monopoly money.
D. have lunch with friends.
Please format your response in the following way:
[Explanation]. Therefore, the answer is answer (label).
Ensure the final sentence includes the answer followed by the
label in parentheses.
Answer and Reasoning: 1. **Analyzes the question**: The
question presents a scenario where a person has identified that
```

---

```
starting to save money is essential for their future,
particularly to afford a vacation at the end of the year. The
goal is to find a way to save money effectively.
2. **Considers the key concepts involved**: The key concepts
here include financial management, expense control, and saving
for future goals. The vacation implies a need for sufficient
funds, which means the person must prioritize actions that lead
to financial growth or reduction of expenses.
3. **Evaluates why the correct answer is right**: Option (C),
\"buy less with monopoly money,\" aligns perfectly with the
goal of saving money. By buying less, the person can apply the
saved money to their vacation fund, ensuring they have enough
resources to cover their expenses and more. This choice
represents a practical and logical approach to financial
management.
4. **Explains why other options are incorrect**:
   - (A) \"make more phone calls\": This option does not save
   money. In fact, making more phone calls could increase
   expenses rather than saving money.
   - (B) \"quit eating lunch out\": While this may save some
   money, it is not a sustainable long-term strategy. Eating
   out is a necessary expense for many people, and quitting it
   could lead to financial hardship.
   - (D) \"have lunch with friends\": Similar to option B, this
   does not contribute to financial saving. It may also create
   additional expense if one does not have the means to support
   it.
5. **Conclusion**: Given the need for financial reserves to
support future goals, the correct answer is clearly (C) buy
less with monopoly money. This option promotes financial growth
through careful expense management.
Therefore, the answer is: (C) buy less with monopoly money.
```

## D.4 MATH

### MATH Example

```
Solve the following math problem step by step. Ensure all
reasoning is shown clearly in LaTeX format, and the final
answer should be enclosed using the LaTeX command \boxed{} to
display it in a box.
Question: What is the positive difference between 120% of 30
and 130% of 20?
Solution: 120% of 30 is $1.2 \\times 30 = 36$ and 130% of 20 is
$1.3 \\times 20 = 26$.
The positive difference between these two values is $36 - 26 =
10$.
So the answer is $10$.
The final answer is: \\boxed{10}
```

