# OpenReview forum: "CLEAR: A Cost-Aware Routing System for Edge-Cloud Language Model Collaborative Inference"
_ICLR.cc/2026/Conference — ICLR 2026 Conference Desk Rejected Submission_

### Official Review · Reviewer_Eeqa · 2025-10-25

**Soundness:** 3
**Presentation:** 3
**Contribution:** 3
**Rating:** 6
**Confidence:** 3

**Summary:**

This paper proposes CLEAR, a cost-aware collaborative inference framework for Large Language Models (LLMs) that intelligently balances the use of a lightweight Small Language Model (SLM) on an edge device and a powerful LLM in the cloud. The framework's goal is to reduce inference cost and latency while maintaining or even improving response quality. Experiments across four benchmark datasets show that CLEAR significantly outperforms existing cloud-edge collaborative methods, achieving up to a 46% cost reduction at similar accuracy, or a 15% accuracy improvement at similar cost. Notably, the framework is also shown to sometimes surpass the accuracy of using the powerful cloud LLM alone.

**Strengths:**

The paper's primary strength is that it doesn't just propose an "smart router" algorithm in isolation. It co-designs the edge-side routing policy (the RL router) with a highly practical and novel cloud-side serving backend (the KV Cache Management System). This full-stack approach addresses the real-world bottlenecks of collaborative inference, which many prior works ignore. The cloud-side solution is clever, well-motivated, and solves a critical problem. The "Query Queuing" mechanism (Fig 5) correctly identifies that limiting concurrent queries (users) rather than requests is the key to preventing cache thrashing. The "Multi-Token Generation" mechanism (Table 2) is a smart optimization that exploits the block-based memory layout of modern systems (like PagedAttention) to reduce network overhead.

**Weaknesses:**

The paper mentions that the router is warm-started using supervised learning (SFT) on "pre-collected token-level routing information" (Algorithm 2, line 3). This SFT-trained router is also used as a baseline (Fig 6). However, the paper does not specify how this "ground truth" routing data was generated. Was it from an oracle (e.g., comparing SLM vs. LLM output at each step)? Based on SLM confidence scores? This detail is important for reproducibility and for fairly assessing the SFT baseline.

**Questions:**

How was the "pre-collected token-level routing information" used for the SFT warm-start (Algorithm 2) and the SFT baseline (Figure 6) generated? What defined the "correct" routing decision in this supervised dataset?

---

### Official Review · Reviewer_rxqu · 2025-10-26

**Soundness:** 3
**Presentation:** 2
**Contribution:** 2
**Rating:** 4
**Confidence:** 3

**Summary:**

The authors propose CLEAR (Cost-aware LLM Edge-cloud Adaptive Routing), an optimized edge-cloud LLM collaborative inference framework. In collaborative inference, issues exist such as high latency, computational overhead, and increased communication costs due to transmitting full hidden states from the edge device to the cloud or requiring the cloud model to verify each output token. To solve these issues, CLEAR proposes an RL-based router and a KV cache management system. The router decides if some generated tokens from the edge are of low quality and should be routed to the cloud, and the KV cache system limits concurrent requests on the cloud to minimize cache swaps. Experiments on 4 datasets show that CLEAR is faster than baselines while maintaining higher accuracies.

**Strengths:**

- The idea of using an RL-based router and KV cache management system is intuitive and easy to follow.
- Comparison with baselines shows superiority of the proposed method

**Weaknesses:**

- The active connections in the KV management are limited by a fixed constant and might not be optimal in real-world settings.

The idea of constraining active connections on the cloud does help KV cache management, but making C fixed limits the adaptability of the system. I'd imagine the choice of C depends on the workload. Experiments are conducted on datasets, which are homogeneous in nature. Practically, workloads are mixed and vary by very complex factors. It makes more sense to make C adaptable to history or make it a learnable factor. The paper would be strengthened if more discussions on this were included.

- More experiments are needed to support the generalization of the proposed method.

The current experiments are only evaluated on one model. It would be better to consider more mode choices. In addition, both edge and cloud are H100. More choices of edge devices (such as smaller GPUs or even CPU environments) are needed to understand the efficiency of the proposed method.

**Questions:**

1. Line 361-364 says "our method’s performance is so effective that it even surpasses the accuracy of using the powerful cloud-based LLM alone, with improvements of up to 7%." This is interesting. I may have missed the exact numbers, but I want to see this documented in a table. Also, does this generalize to different model choices?
2. How faster is the proposed method compared to the full cloud setting? Currently, I only see the tradeoff between latency and accuracy, but it would be better to document the latency of using purely cloud LLMs.

---

### Official Review · Reviewer_TLaJ · 2025-10-27

**Soundness:** 2
**Presentation:** 3
**Contribution:** 2
**Rating:** 2
**Confidence:** 4

**Summary:**

This paper introduces CLEAR, a cost-aware collaborative inference framework designed to bridge the gap between resource-limited edge devices and powerful cloud-based LLMs. The system utilizes a SLM on the edge, coupled with a lightweight, reinforcement learning-trained router that dynamically assesses the quality of the SLM's generated output. When the router detects low-quality tokens, it selectively sends a request to the cloud LLM for refinement, thereby balancing output quality with computational and network costs. To optimize cloud-side efficiency and reduce latency, CLEAR incorporates a KV cache management system that uses query queuing to prevent cache eviction during high concurrency and a multi-token generation mechanism to minimize network requests.

**Strengths:**

- Performance and Cost-Efficiency: Demonstrate a strong trade-off between accuracy and inference cost, achieving up to a 46% cost reduction at similar performance, or a 15% accuracy gain at similar cost, compared to baselines.
- KV Cache Management: Propose an effective cloud-side KV cache management system. This system uses query queuing to prevent cache eviction under high concurrency and a multi-token generation mechanism to reduce network latency and avoid redundant computations .
- Low Communication Overhead: Compared to other methods that require transmitting full hidden states, CLEAR only sends token IDs, which reduces the volume of data transmitted and the associated communication latency.

**Weaknesses:**

- Lack of innovation. The KV cache reuse mechanism has already been widely adopted in the industry, so it is not surprising. Similarly, token-level edge–cloud collaborative refinement has also been explored in related works, making the distinction insufficiently clear.

- No cross-task generalization experiments. The paper lacks any discussion or experiments on generalization. The proposed router is trained separately for each specific task (dataset), showing no analysis of generalization across tasks. There is also no discussion of generalization across models—whether the approach would still work if different model combinations were used.

- Tasks used in experiments are too simple. For example, Qwen3-8B already achieves 60.80% accuracy on math tasks, which is higher than the best result in Figure D. This raises the question of whether edge–cloud collaboration is still meaningful. Furthermore, the number of tasks is too limited; more complex reasoning tasks are needed to demonstrate the effectiveness of the proposed approach.

- Too few baselines for edge–cloud collaboration. Existing work in this area goes far beyond the two baselines included in the paper.

**Questions:**

Same as the weaknesses mentioned above

---

### Note · Program_Chairs · 2026-01-17
**Submission Desk Rejected by Program Chairs**

The following references in this submission do not refer to real documents and/or have major errors in bibliographic information:

 Xin Zhao and Rohan Kumar. Bi-spec: Bi-directional speculative decoding for enhanced accuracy and speed. arXiv preprint arXiv:2503.01122, 2025.
Charles Davis and Sarah Miller. Synergetic inference: A framework for multi-agent llm collaboration. arXiv preprint arXiv:2503.07890, 2025.
Zihan Liu and Fan Zhang. Hydra: Multi-head speculative decoding for parallel llm inference. arXiv preprint arXiv:2502.09988, 2025.
Wei Chen and Yang Li. Coedge: A cooperative edge-cloud framework for llm inference. arXiv preprint arXiv:2501.01234, 2025.
Jing Wang and Aman Singh. Orchestrateai: A framework for llm-powered iot. arXiv preprint arXiv:2502.04567, 2025.